# Effects of Pulsed Electric Field Pretreatment on Black Tea Processing and Its Impact on Cold-Brewed Tea

**DOI:** 10.3390/foods13010164

**Published:** 2024-01-03

**Authors:** Hyunho Yeo, Si-Yeon Kim, Hafiz Muhammad Shahbaz, Se-Ho Jeong, Hye-In Ju, Ji-Hee Jeon, Dong-Un Lee

**Affiliations:** 1Department of Food Science and Technology, Chung-Ang University, Anseong 17546, Republic of Korea; yhh0437@ottogi.co.kr (H.Y.); asdfisiof1@cau.ac.kr (S.-Y.K.); mrr003@cau.ac.kr (S.-H.J.); dldkqm@cau.ac.kr (H.-I.J.); wjswlgml901@cau.ac.kr (J.-H.J.); 2R&D Center, Ottogi Co., Ltd., Anyang 14060, Republic of Korea; 3Department of Food Science and Human Nutrition, University of Veterinary and Animal Sciences, Lahore 54000, Pakistan; muhammad.shahbaz@uvas.edu.pk

**Keywords:** PEF, cold-brewed black tea, withering, drying kinetics, sensory evaluation

## Abstract

This study applied pulsed electric fields (PEFs) to accelerate the withering and drying processes during cold-brewed black tea production. PEF pretreatment was administered at 1.0, 1.5, and 2.0 kV/cm electric field strengths, combined with varying withering times from 8 to 12 hr. During the 12-hour withering process, the redness value (a*) and total color change (∆E) of PEF-treated leaves significantly increased (*p* < 0.05). Furthermore, the homogenous redness of tea leaves during fermentation depended on the PEF strength applied. In addition, PEF pretreatment remarkably reduced the drying time, up to a 50% reduction at a 2.0 kV/cm field strength. Additionally, the 2.0 kV/cm PEF-pretreated black tea exhibited a notable 42% increase in theaflavin (TF) content and a 54% increase in thearubigin (TR) content. Sensory evaluation scores were highest for black tea that received PEF pretreatment at 2.0 kV/cm. These findings highlight the significant potential of PEFs in enhancing the efficiency of withering and drying processes while positively impacting the physicochemical and sensory properties of cold-brewed black tea.

## 1. Introduction

Black tea is derived from the fermentation of green tea leaves, in which polyphenol oxidase enzymes convert catechins into theaflavins (TFs) and thearubigins (TRs). These compounds impart the characteristic flavor and health benefits associated with black tea [1]. Traditional black tea processing involves plucking, withering, rolling, fermentation, and drying, with the withering and fermentation stages contributing significantly to developing the tea’s aroma [2].

In particular, withering is a time-consuming process that reduces the moisture content of fresh tea leaves from approximately 75% to 55–65%. During withering, chemical compounds within the leaves break down, altering flavor, texture, and other chemical properties [3]. Manufacturing black tea requires a lengthy processing time, resulting in low productivity and high energy consumption, necessitating improvements for this stage of production.

Pulsed electric fields (PEFs) are a non-thermal technology that delivers short bursts of high-voltage electrical pulses, inducing electroporation and augmenting cellular membrane permeability [4]. PEFs have been utilized in various food processing applications, including enhancing drying and rehydration efficiency [5] and improving the extraction yield of bioactive compounds [6]. Regarding tea production, PEFs have been used to accelerate the drying process of tea [7] and enhance polyphenol extraction from tea leaves [8]. Additionally, studies have demonstrated that PEFs can enrich the tea’s quality and sensory properties [9].

Cold brew is a popular brewing method for tea, known for preserving heat-sensitive bio-active compounds, such as the antioxidants and anti-inflammatory agents that contribute to its nutritive value and potent aroma [10]. However, cold brewing has limitations, including lower extraction efficiency and higher production costs than traditional hot water brewing methods [11]. Researchers have explored various techniques to enhance the extraction efficiency and reduce the processing time for cold brew tea, including ultrasound-assisted extraction [12] and combined technologies, like ultrasound with high hydrostatic pressure and puffing drying [13]. Despite these efforts, research on applying cold brewing technologies to black tea processing remains insufficient.

Despite the promising applications of PEFs in the tea industry, no studies have investigated their effects on the processing of cold-brewed black tea. Therefore, this research explores the potential benefits of PEFs in reducing withering and drying times and improving cold-brewed black tea’s physicochemical properties, including color, browning index, theaflavin and thearubigin contents, and sensory properties. This research provides valuable scientific data on the effects of pulsed electric fields in manufacturing cold-brewed black tea.

## 2. Materials and Methods

### 2.1. Materials

Fresh tea leaves (*Camellia sinensis* var. *sinensis*) were harvested between the end of August and the middle of September 2022 from a local farm in Hadong-gun, Gyeongsangnam-do, Republic of Korea. Upon harvest, the tea leaves were immediately transported to the laboratory and carefully washed with distilled water to remove any surface impurities. The samples were then stored at 4 °C for a maximum of 48 h until experimentation. Tea leaves measuring 4–5 cm in length and 1–2 cm in width were selected for the experiments.

### 2.2. PEF Pretreatment

Pulsed electric field (PEF) pretreatment was accomplished using a 5 kW pulse generator (HVP-5, DIL, Quakenbrück, Germany) with a batch-type chamber maintained at room temperature. The batch chamber comprised two parallel steel electrodes with an electrode distance set at 80 mm. Fresh tea leaves (2 g) were placed into the chamber with 200 mL of tap water at 20 °C. The leaves were completely submerged in tap water and placed so as not to come into direct contact with the electrodes. PEF pretreatment was conducted at 1.0, 1.5, and 2.0 kV/cm electric field strengths using a 20 μs fixed pulse width (τ), a 50 Hz frequency, and a pulse number (n) of 500. The samples were designated as Control (untreated), PEF1.0, PEF1.5, and PEF2.0.

### 2.3. Manufacturing of Black Tea and Tea Sample Preparation

After PEF treatment, the tea leaves’ surface moisture was removed using paper towels and the leaves were then processed into black tea following a method by Pou [14], with slight modifications (Appendix A). The withering process was conducted at 23 °C and 60% relative humidity until the moisture content decreased to 55–65%. The withering time (WT) was categorized into three intervals of 8, 10, and 12 h and samples were designated as WT8h, WT10h, and WT12h, respectively. Next, the tea leaves were fermented in a 10 cm thick layer at 30 °C and 95% relative humidity for 1.5 h. After fermentation, the samples were dried in a hot air oven (SFC-203, Shinsaeng, Paju-si, Republic of Korea) at 70 °C for 1.5 h, followed by further drying until the moisture content dropped below 8%. The dried tea leaves were then cooled to room temperature. For the preparation of cold-brewed black tea, 2 g of dried samples were infused with 120 mL of distilled water at 20 °C for 12 h. The resulting infusion was filtered using 0.45 μm PTFE syringe filters and maintained at 4 °C until further experimentation.

### 2.4. Temperature Change, Electric Conductivity, and Ion Leaching

Temperature changes and electrical conductivity were measured before and after PEF treatment to evaluate its effects on tap water. The electrical conductivity of tap water was determined with a conductivity meter (CM-21P, TOA-DKK, Tokyo, Japan). Ion leaching measurements were conducted to indirectly estimate the increase in the membrane permeability of tea leaves [15]. Following PEF treatment, 4 g of samples were immediately immersed in 100 mL of distilled water at 20 °C. The electrical conductivity of the water was measured every hour for 6 h at room temperature using a conductivity meter (CM-21P, TOA-DKK, Japan).

### 2.5. Determination of Fermentation Degree

The fermentation degree of the samples was assessed through photographs taken at 1-h intervals during the 8-, 10-, and 12-h withering periods. The red area of six fermented samples under each condition was calculated using ImageJ software (version 1.53, National Institute of Health, Bethesda, MD, USA) to determine the fermentation degree. The fermentation degree was expressed as the ratio of the red area to the whole leaf area using Equation (1), as described by Matsunaga, et al. [16], with slight modifications. Appropriate HSB color space and thresholds were applied to select and convert the red and whole areas into pixels:(1)$$Fermentation\;degree\left(\%\right)=\frac{\mathrm{Red}\;\mathrm{area}\;\mathrm{of}\;\mathrm{tea}\;\mathrm{leaves}}{\mathrm{Whole}\;\mathrm{area}\;\mathrm{of}\;\mathrm{tea}\;\mathrm{leaves}}$$

### 2.6. Color Measurement

The colors of the samples during the withering process and the cold-brewed black tea infusion were measured using a colorimeter (CR-400, Minolta, Osaka, Japan). After calibration with standard black and white plates, the CIELAB value determined the tea color. The center area of each sample was measured every hour during the withering process and 1 mL of the cold-brewed black tea infusion was measured after 12 h of infusion. The total color change (∆*E*) and browning index (*BI*) were calculated using Equations (2) and (3), as described by Matsunaga, Ogawa, Taguchi-Shiobara, Ishimoto, Matsunaga, and Habu [16]:(2)$$∆E=\sqrt{{\left(L_{0}^{*}-L^{*}\right)}^{2}+{\left(a_{0}^{*}-a^{*}\right)}^{2}+{\left(b_{0}^{*}-b^{*}\right)}^{2}}$$
where $L_{0}^{*}$, $a_{0}^{*}$, and $b_{0}^{*}$ are the initial color values and *L**, *a**, *b** are the lightness, redness, and yellowness of tea leaves:(3)$$BI=\frac{[100\left(x-\;0.31\right)]}{0.17}$$
where x = (a + 1.75L)/(5.645L + a − 3.012b).

### 2.7. Moisture Content and Ratio

The moisture content (*MC*) fluctuations during the withering process were monitored by weighing 4 g of samples every 0.5 h based on weight changes. The initial moisture content of fresh tea leaves (75.2% wet basis) was determined by drying 4 g of fresh tea leaves in a drying oven (VS-1202D2, Vision Science Co., Daejeon, Republic of Korea) for 48 h at 105 °C. The withering time was classified into three intervals (8, 10, and 12 h) and terminated when the moisture content fell within 55 to 65%. The moisture content at each time point was calculated using Equation (4):(4)$$MC(\%)\;=\frac{W_{t}-W_{f}}{W_{i}}\;\times \;100$$
where MC is the moisture content (wet basis), W_i_ is the initial weight of the sample, W_t_ is the weight of the sample at a specific time, and W_f_ is the final weight of the sample after the drying process.

The moisture ratio (MR) shift during the drying process was measured by weighing 4 g of samples every 10 min based on weight changes. The moisture content was calculated on a dry basis to express the moisture ratio. The moisture ratio at each time point was determined using Equation (5):(5)$$MR=\frac{M_{t}-M_{e}}{M_{0}-M_{e}}$$
where MR is the moisture ratio (g water/g dry matter), M_t_ is the moisture content of the sample at time t, M_0_ is the initial moisture content, and M_e_ is the equilibrium moisture content. M_e_ was determined by measuring at a constant weight point [17].

### 2.8. pH, Total Soluble Solids (TSS), and Turbidity

The physical characteristics of the cold-brewed black tea infusion were determined by measuring the pH, total soluble solids (TSS), and turbidity. The pH value was measured using a pH meter (Sevencompact TM S210, Mettler Toledo, Greifensee, Switzerland) and the total soluble solids (TSS) were calculated using a refractometer (Pocket PAL-1, ATAGO, Tokyo, Japan) [18]. Turbidity was measured as absorbance at 680 nm using a spectrophotometer (Genesys 20, Thermo Scientific, Waltham, MA, USA) with distilled water as the blank. All experiments were performed in triplicate at 25 °C.

### 2.9. Determination of Theaflavin (TF) and Thearubigin (TR) Contents

The chemical characteristics of the cold-brewed black tea were assessed by determining TF and TR contents, which were estimated following the method described by Ullah [19]. First, 12 mL of the samples were mixed with 6 mL of a 1% (*w*/*v*) aqueous solution of anhydrous disodium hydrogen phosphate and extracted with 10 mL of ethyl acetate by shaking for 1 min. The ethyl acetate layer (TF fraction) was collected and 10 mL of the TF extract was diluted to 25 mL with 70% methanol and designated as Extract 1 (E_1_). Next, 1 mL of 10% (*w*/*v*) aqueous oxalic acid and 8 mL of distilled water were added to 1 mL of tea infusion and the volume was adjusted to 25 mL with 70% methanol to create Extract 2 (E_2_). The E_1_ and E_2_ optical densities were obtained at 380 nm after applying suitable corrections for the extract strength and actual volumes used. TF and TR contents of the tea infusion were calculated using Equations (6) and (7), respectively:(6)$$TF\left(\%\right)=2.25\times {OD}_{E_{1}}$$
(7)$$TR\left(\%\right)=7.06\times (4{OD}_{E_{2}}-{OD}_{E_{1}})$$
where ${OD}_{E_{1}}$ is the optical density of E_1_ and ${OD}_{E_{2}}$ is the optical density of E_2_ at 380 nm.

### 2.10. Sensory Evaluation

The sensory evaluation of black tea was approved by the Chung-Ang University Institutional Review Board (approval number: 1041078-20221107-HR-008). The evaluation was performed using WT12h samples. For sample preparation, 14 g of black tea leaves was infused with 840 mL of water at 20 °C for 12 h and then poured into a 1 L tea-tasting bowl. In total, 15 mL of each sample was evaluated at different areas to minimize olfactory fatigue. A panel of 40 evaluators of both genders, aged between 20 and 30 years, participated in the sensory evaluation. The panelists majored in Food Science. Four attributes were evaluated (flavor, taste, color, and overall acceptability) using a 9-point hedonic scale (1 = dislike extremely, 9 = like extremely), as described by Adnan, et al. [20], with slight modifications.

### 2.11. Statistical Analysis

The data are presented as the means of three measurements ± standard deviation (SD). The data were analyzed through ANOVA and Duncan’s multiple range comparison tests using SPSS ver. 26.0 software (IBM, Armonk, NY, USA). Statistical significance was determined at *p* < 0.05.

## 3. Results and Discussion

### 3.1. PEF Effects on Temperature, Electric Conductivity, and Ion Leaching

Pulsed electric field (PEF) effects on the temperature, electric conductivity, and ion leaching during black tea processing were investigated. As PEF strength increased, the temperature of the solution after PEF treatment also increased slightly. Specifically, PEF2.0 exhibited the highest temperature of 21.4 °C compared to PEF1.0 (20.5 °C) (Figure 1A). This observation indicates that PEF treatment slightly impacted the temperature of the tap water as media as higher field strengths elevated temperature. Figure 1B illustrates electric conductivity changes in the sample after PEF treatment; the initial electric conductivity was 19.87 mS/m. After PEF treatment, the PEF1.0, PEF1.5, and PEF2.0 electric conductivity values were 20.03, 20.34, and 20.78 mS/m, respectively. After 6 h, the Control, PEF1.0, PEF1.5, and PEF2.0 electric conductivity values were 5.87, 7.86, 17.21, and 20.95 mS/m, respectively (Figure 1C). These results indicate that PEF treatment altered the tea solution’s electric conductivity, with higher field strengths increasing electric conductivity. The conductivity could represent a degree of cell membrane disruption and was dependent on PEF strength [21]. The conventional black tea manufacturing process needs the rolling procedure, which damages the leaves for the fermentation [22]. The PEF treatment, which selectively damages the cell membrane, could be an alternative method for the rolling. The minimum and maximum energy for the rolling process was 360 to 720 MJ/t for processing black tea [23]. The energy requirements in the current study were lower than those in the previous study. These findings suggest that energy consumption in the conventional black tea manufacturing process was low with PEF pretreatment rather than the rolling process.

### 3.2. Color Changes during the Withering Process

Tea leaf colors were evaluated during the withering process and significant differences were observed in the *a** value among samples. As withering time increased, the difference between the Control and the sample treated with 2.0 kV/cm increased, with the highest value observed in the 2.0 kV/cm sample (−0.82 ± 0.62) after 12 h. However, L* and b* values decreased as PEF strength increased. Specifically, the 2.0 kV/cm sample presented the lowest *L** value (32.24 ± 0.32) while the Control exhibited the highest (38.02 ± 0.56). The Control, 1.0 kV/cm, 1.5 kV/cm, and 2.0 kV/cm *b** values were 20.21 ± 0.46, 18.78 ± 1.20, 17.13 ± 0.97, and 14.82 ± 0.47, respectively (Figure 2). Changes in a* values also significantly increased the total color change (∆E) during the withering process. The Control, PEF1.0, PEF1.5, and PEF2.0 total color change values were 3.28, 6.78, 13.89, and 18.41, respectively (Figure 2). These results indicate that PEF pretreatment during withering impacted the color development of the black tea leaves, with higher PEF strengths effectuating more pronounced color changes. The observed color changes could be attributed to catechin oxidation in the tea leaves. Catechins in the vacuole interact with polyphenol oxidase (PPO) and peroxidase (POD) in the cytoplasm, transitioning the color from green to brownish red [24]. Notably, the withering process in this study induced color changes, likely attributed to PEF pretreatment causing the disruption of the cell membrane and tonoplast. These intracellular changes facilitated catechins and PPO/POD interactions, thereby initiating oxidation in tea leaves [25].

### 3.3. Fermentation Degree Assessment

Table 1 and Figure 3 present the fermentation degree of tea. At WT8h (8-hour withering time), WT10h (10-hour withering time), and WT12h (12-hour withering time), the fermentation degree increased as PEF strength increased for all samples (*p* < 0.05). Notably, the fermentation degree of leaves at the same withering time displayed different aspects. WT8h, WT10h, and WT12h indicated no significant differences at the Control and PEF2.0 field strengths. However, the fermentation degree significantly increased with higher field strengths at PEF1.0 and PEF1.5 (*p* < 0.05). The “red stain” observed in the fermented tea leaves is a color change characteristic of the dynamic pigment transition from polyphenols to theaflavin (TF) and thearubigin (TR). Mottled and uneven red stains negatively affect the black tea quality [26]. However, in this study, as PEF strength increased, the red stain in the tea leaves became more apparent. This phenomenon is caused by the PEF energy homogenously affecting the cells in leaf tissues. Higher PEF strengths beget a greater ratio of damaged cells, increasing the fermentation degree in tea leaves. Moreover, as the withering time increased from 8 to 12 h, the fermentation degree intensified under the same field strength conditions. PEF pretreatment at 1.0 kV/cm and 1.5 kV/cm accelerated tea leaf fermentation as withering time increased while pretreatment at 2.0 kV/cm was stronger than that at 12 h.

### 3.4. Changes in Moisture Content and Ratios

#### 3.4.1. Moisture Content

The moisture content of the tea leaves during the withering and drying processes was investigated. The initial moisture content of the fresh sample was 75.02 ± 0.10%. After 12 h of withering, the Control, 1.0 kV/cm, 1.5 kV/cm, and 2.0 kV/cm moisture contents were 63.44 ± 0.10%, 62.31 ± 0.12%, 60.76 ± 0.15%, and 59.22 ± 0.10%, respectively (Figure 4). When manufacturing high-quality black tea, the appropriate moisture content for tea leaves after withering is 55–65%. In this study, the 2.0 kV/cm treatment only required 6 h to reach a 65% moisture content, compared to the 9 h needed for the Control. This phenomenon was related to the enhanced membrane permeability induced by PEF treatment. The PEF-treated leaves gained porosity of the cell membrane due to the PEFs not affecting the cell wall structure [15,27]. This porosity structure could facilitate dehydration. Previous studies reported that PEF applications significantly decreased the moisture content in basil leaves [28].

#### 3.4.2. Moisture Ratio

The moisture ratio (MR) during hot air drying at 70 °C and the time taken for each sample to reach a below 8% moisture content was recorded. The results indicated that PEF pretreatment significantly reduced the drying time of the tea leaves. PEF1.0, PEF1.5, and PEF2.0 samples under the WT8h condition required 70, 60, and 50 min, respectively, to reach dry weight compared to 80 min for the Control. In the WT10h condition, PEF1.0, PEF1.5, and PEF2.0 samples under the WT10h conditions required 60, 60, and 40 min, respectively, compared to the 80 min for the Control. Similarly, PEF1.0, PEF1.5, and PEF2.0 samples under the WT12h conditions required 60, 50, and 40 min compared to the 80 min for the Control. These results indicate that PEF pretreatment significantly reduced the drying time of tea leaves, with higher PEF strengths resulting in a more rapid MR reduction. The PEF treatment could induce the increase in cell membrane porosity via the disintegration of the tonoplast and plasma membrane while maintaining the intact cell wall structure [15,28]. The porosified cells can easily transport intracellular moisture. The intact structure could also contribute to the acceleration of the drying rate by preventing the shrinkage of the cell wall. Similar results were reported in previous studies involving PEF treatment on basil leaves and red pepper [29,30]. Thus, the PEF pretreatment induced membrane disruption and increased the mass transfer of water during the drying process, prompting faster drying times.

### 3.5. Physical Characteristics

The physical characteristics of the cold brew black tea infusion were assessed, including pH, total soluble solids (TSS), turbidity values, and color. The pH of the black tea infusion was neutral, ranging from 5.26 ± 0.03 to 5.55 ± 0.08 in all samples. As PEF strength increased, the pH value decreased slightly; however, no significant differences were observed, except for WT12h-PEF2.0 (5.26 ± 0.03 pH) (Table 1). Neither the withering time nor the PEF strength significantly affected the pH of the black tea infusion. At WT12h, the Control, PEF1.0, PEF1.5, and PEF2.0 total soluble solid (TSS) values were 0.33 ± 0.06, 0.33 ± 0.06, 0.43 ± 0.06, and 0.67 ± 0.06 °Brix, respectively (Table 1). Notably, the WT12h-PEF2.0 sample achieved the highest TSS value. The turbidity of the black tea infusion also impacted its quality. At WT12h, the Control, PEF1.0, PEF1.5, and PEF2.0 turbidity (%T) values were 98.17 ± 0.15, 97.77 ± 0.25, 97.13 ± 0.31, and 95.10 ± 0.44%T, respectively (Table 1). These results corroborate previous studies observing that PEF pretreatment increased the soluble solids content in chicory [31]. The enhanced cell membrane permeability induced by PEF treatment accelerated the leaching of soluble solids from the tea leaves. Thus, higher PEF strengths increased cell membrane permeability, accelerating soluble solid leaching in leaves.

The CIELAB values detailed in Table 1 reveal the color changes of cold brew black tea relative to theaflavin (TF) and thearubigin (TR) content. Also, the corresponding appearances of the teas are presented in Figure 3C. Notably, the L* value declined with field strength, from 88.28 ± 0.06 to 85.38 ± 0.03. The a* value reduced and the b* value increased, most notably between 13.16 ± 0.08 and 24.06 0.11 for WT12h-CON and 13.16 ± 0.08 and 24.0 ± 0.11 for WT12h-PEF2.0. Furthermore, the Browning Index (BI) increased alongside field strength and withering time. TF (which contributes to the bright and orange-yellow shade) and TR (responsible for the red-brown hue) contents elevated with rising field strength. Specifically, regarding a field strength jump from untreated to PEF2.0 at WT12h, TF values altered from 0.38 ± 0.02% to 0.54 ± 0.01% and TR values shifted from 4.00 ± 0.09% to 6.14 ± 0.12%. Interestingly, despite an increase in TF and TR, a* values decreased. This behavior suggests that TF oxidizes to TR as the withering period extends, resulting in a deeper-colored cold brew black tea, especially at higher field strengths [2].

### 3.6. Sensory Evaluation

The sensory evaluation of the cold brew black tea infusion was conducted by 40 evaluators using a nine-point hedonic scale to assess flavor, taste, color, and overall acceptability. The results indicated that PEF-pretreated cold brew black tea samples performed better concerning color, flavor, taste, and overall acceptability than the untreated Control samples (Table 2). All attributes exhibited significant differences (*p* < 0.05), with color displaying the most pronounced differences between the Control (4.73 ± 0.58) and PEF2.0 (8.37 ± 0.56). In contrast, flavor showed the slightest differences between the Control (5.63 ± 0.49) and PEF2.0 (7.83 ± 0.75) (Table 2). Previous studies reported that high-quality black tea is characterized by more prominent red and yellow colors, which are associated with higher theaflavin (TF) concentrations [2,32]. Additionally, high TF and thearubigin (TR) concentrations positively influenced the taste and color of black tea. In this study, PEF2.0 achieved higher TF and TR contents than the Control, resulting in higher sensory evaluation scores for color, flavor, taste, and overall acceptability.

## 4. Conclusions

PEF pretreatment positively impacted the withering, fermentation, and drying processes during black tea manufacturing. Notably, as the PEF strength increased, withering time decreased, fermentation degree increased, and total drying time was reduced. PEF pretreatment positively affected the physicochemical characteristics of the cold brew black tea infusion, such as soluble solids, theaflavin, and thearubigin. Sensory evaluation results indicated that PEF-pretreated cold brew black tea samples performed better regarding color, flavor, taste, and overall acceptability than untreated Control samples. Overall, PEF pretreatment is a promising method to enhance black tea manufacturing productivity and the physicochemical characteristics of cold brew black tea.

## Figures and Tables

**Figure 1 foods-13-00164-f001:**
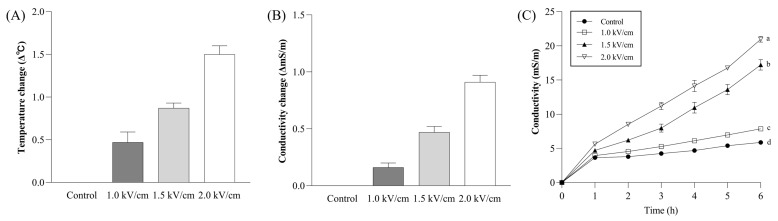
Effect of PEF treatment on the (**A**) temperature change, (**B**) conductivity change, and (**C**) ion release from tea leaves. The meanings of the symbols in the figure: ●—Control, □—1.0 kV/cm, ▲—1.5 kV/cm, ▽—2.0 kV/cm. All values are expressed by mean ± SD (n = 3). Values with different letters (a–d) are significantly different (*p* < 0.05), as determined by Duncan’s test.

**Figure 2 foods-13-00164-f002:**
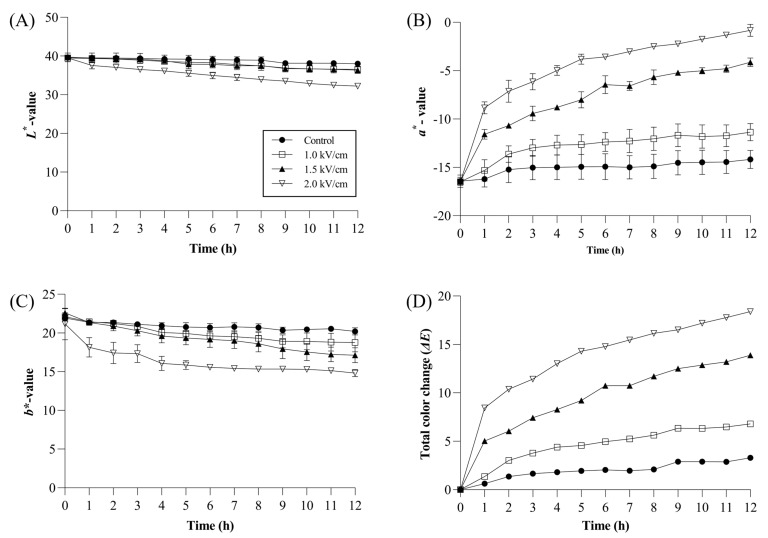
CIELAB values and the total color change of the Control and PEF-treated tea leaves during the withering process. (**A**) *L** value, (**B**) *a** value, (**C**) *b** value, (**D**) total color change *(*∆*E*). The meanings of the symbols in the figure: ●—Control, □—1.0 kV/cm, ▲—1.5 kV/cm, ▽—2.0 kV/cm. All values are expressed by mean ± SD (n = 3).

**Figure 3 foods-13-00164-f003:**
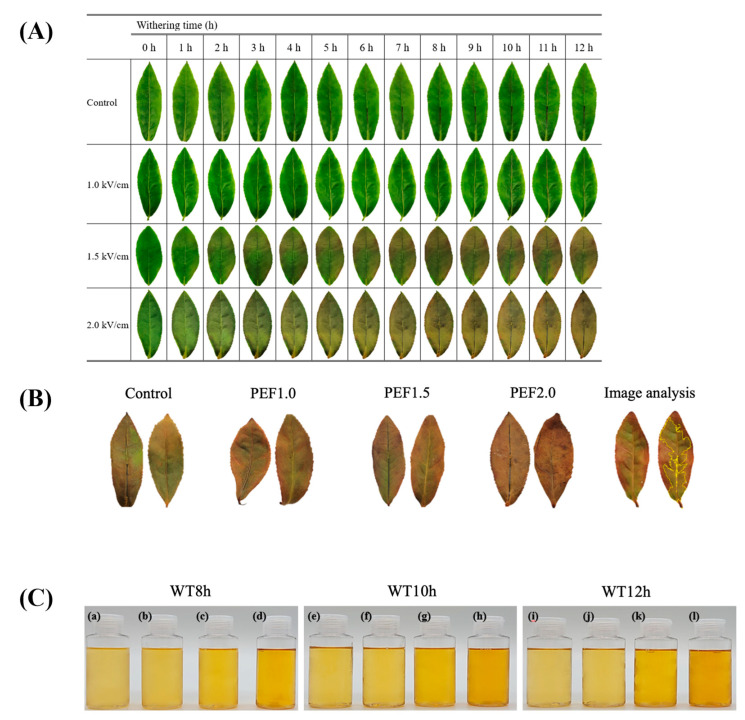
Photograph of tea leaves during the withering process (**A**), after fermentation (**B**), and after cold brew black tea extraction (**C**). (**a**,**e**,**i**): Control, (**b**,**f**,**j**): PEF1.0, (**c**,**g**,**k**): PEF1.5, (**d**,**h**,**l**): PEF2.0.

**Figure 4 foods-13-00164-f004:**
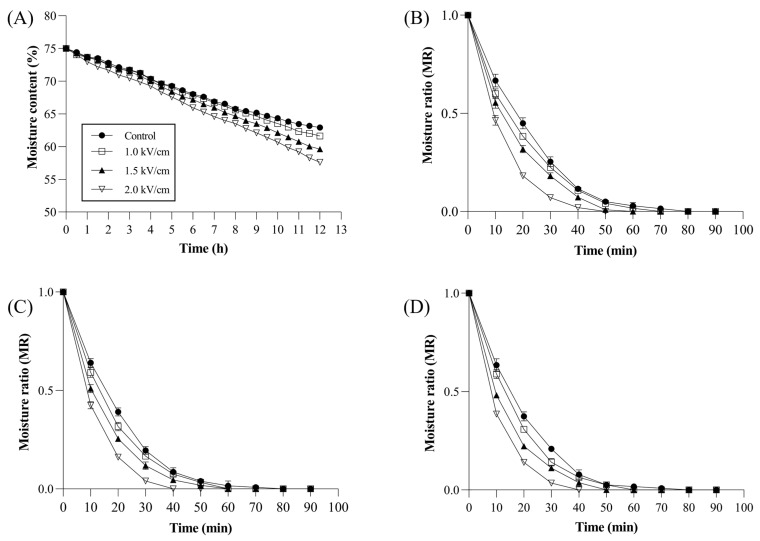
Effect of PEF treatment on the moisture content after the withering process (**A**) and moisture ratio during the drying process; (**B**): WT8 h; (**C**): WT10h; (**D**): WT12h. The meanings of the symbols in the figure: ●—Control, □—1.0 kV/cm, ▲—1.5 kV/cm, ▽—2.0 kV/cm. All values are expressed by mean ± SD (n = 3).

**Table 1 foods-13-00164-t001:** Physical characteristics, color, browning index, and theaflavin and thearubigin contents of cold brew black tea.

Sample	Withering Time (h)	Physical Characteristics			Color				Fermentation Degree (%)	Theaflavin (%)	Thearubigin (%)
		pH	TSS (°Brix)	Turbidity (%)	*L**	*a**	*b**	Browing Index (*BI*)			
Control	8	5.55 ± 0.08 ^aA^	0.33 ± 0.06 ^cA^	98.10 ± 0.10 ^aA^	88.28 ± 0.06 ^aA^	−1.56 ± 0.01 ^aA^	12.74 ± 0.10 ^dB^	13.89 ± 0.13 ^dC^	62.44 ± 3.05 ^dA^	0.37 ± 0.01 ^dA^	3.62 ± 0.05 ^dB^
	10	5.54 ± 0.06 ^aA^	0.37 ± 0.06 ^aA^	98.10 ± 0.10 ^aA^	87.60 ± 0.04 ^aB^	−1.65 ± 0.04 ^aA^	12.88 ± 0.04 ^dB^	14.11 ± 0.03 ^dB^	65.76 ± 3.20 ^dA^	0.39 ± 0.01 ^cA^	3.91 ± 0.01 ^cA^
	12	5.46 ± 0.04 ^aA^	0.33 ± 0.06 ^bA^	98.17 ± 0.15 ^aA^	87.98 ± 0.07 ^aC^	−1.62 ± 0.01 ^aA^	13.16 ± 0.08 ^dA^	14.43 ± 0.10 ^dA^	66.87 ± 3.19 ^dA^	0.38 ± 0.02 ^dA^	4.00 ± 0.09 ^dA^
1.0 kV/cm	8	5.49 ± 0.06 ^aA^	0.43 ± 0.06 ^bcA^	98.07 ± 0.06 ^aA^	88.00 ± 0.01 ^bA^	−1.77 ± 0.01 ^bA^	14.34 ± 0.11 ^cA^	15.85 ± 0.14 ^cA^	67.98 ± 1.61 ^cC^	0.40 ± 0.01 ^cA^	4.18 ± 0.09 ^cB^
	10	5.50 ± 0.03 ^aA^	0.47 ± 0.06 ^aA^	98.10 ± 0.10 ^aA^	87.55 ± 0.11 ^aA^	−1.79 ± 0.07 ^bA^	14.32 ± 0.04 ^cA^	15.90 ± 0.13 ^cA^	70.42 ± 2.61 ^cB^	0.41 ± 0.02 ^cA^	4.21 ± 0.09 ^cB^
	12	5.45 ± 0.03 ^aA^	0.33 ± 0.06 ^bA^	97.77 ± 0.25 ^aA^	87.84 ± 0.12 ^bB^	−1.77 ± 0.02 ^bA^	14.26 ± 0.05 ^cA^	15.77 ± 0.05 ^cA^	73.99 ± 2.23 ^cA^	0.42 ± 0.01 ^cA^	4.79 ± 0.08 ^cA^
1.5 kV/cm	8	5.43 ± 0.03 ^aA^	0.53 ± 0.06 ^abA^	97.30 ± 0.44 ^bA^	87.42 ± 0.10 ^cA^	−2.05 ± 0.02 ^cA^	16.93 ± 0.18 ^bC^	19.23 ± 0.25 ^bC^	80.46 ± 1.11 ^bC^	0.46 ± 0.01 ^bA^	4.87 ± 0.17 ^bA^
	10	5.48 ± 0.05 ^aA^	0.43 ± 0.06 ^aA^	97.27 ± 0.68 ^aA^	86.90 ± 0.13 ^bB^	−2.09 ± 0.02 ^cA^	18.27 ± 0.49 ^bB^	21.20 ± 0.72 ^bB^	82.97 ± 1.17 ^bB^	0.46 ± 0.01 ^bA^	4.96 ± 0.36 ^bA^
	12	5.41 ± 0.01 ^aA^	0.43 ± 0.06 ^bA^	97.13 ± 0.31 ^bA^	86.54 ± 0.04 ^cC^	−2.05 ± 0.01 ^cA^	19.69 ± 0.16 ^bA^	23.37 ± 0.24 ^bA^	84.93 ± 1.45 ^bA^	0.46 ± 0.01 ^bA^	5.32 ± 0.27 ^bA^
2.0 kV/cm	8	5.40 ± 0.02 ^aA^	0.63 ± 0.06 ^aA^	95.67 ± 0.42 ^cA^	87.42 ± 0.10 ^cA^	−2.06 ± 0.02 ^cA^	21.51 ± 0.17 ^aC^	26.18 ± 0.27 ^aC^	98.85 ± 0.16 ^aA^	0.51 ± 0.01 ^aA^	5.46 ± 0.04 ^aC^
	10	5.39 ± 0.03 ^aA^	0.60 ± 0.10 ^aA^	95.60 ± 0.53 ^bA^	86.90 ± 0.13 ^bB^	−2.11 ± 0.04 ^cA^	23.34 ± 0.22 ^aB^	29.13 ± 0.35 ^aB^	98.83 ± 0.37 ^aA^	0.53 ± 0.01 ^aA^	5.79 ± 0.19 ^aB^
	12	5.26 ± 0.03 ^bB^	0.67 ± 0.06 ^aA^	95.10 ± 0.44 ^cA^	86.54 ± 0.04 ^cC^	−2.13 ± 0.04 ^dA^	24.06 ± 0.11 ^aA^	30.32 ± 0.21 ^aA^	98.81 ± 0.24 ^aA^	0.54 ± 0.01 ^aA^	6.14 ± 0.12 ^aA^

All values are expressed by mean ± SD (n = 3). Values within different lowercase letters (a–d) in the same column are significantly different (*p* < 0.05), as determined by Duncan’s test. Values within different uppercase letters (A–C) in the same row are significantly different (*p* < 0.05) as determined by Duncan’s test.

**Table 2 foods-13-00164-t002:** Sensory evaluation of cold brew black tea infusion.

Samples	Color	Flavor	Taste	Overall Acceptability
Control	4.73 ± 0.58 ^d^	5.63 ± 0.49 ^d^	5.00 ± 0.37 ^d^	5.13 ± 0.78 ^d^
1.0 kV/cm	5.70 ± 1.09 ^c^	6.17 ± 0.46 ^c^	5.53 ± 0.51 ^c^	5.67 ± 0.66 ^c^
1.5 kV/cm	7.57 ± 0.57 ^b^	6.63 ± 0.72 ^b^	7.77 ± 0.73 ^b^	6.60 ± 0.72 ^b^
2.0 kV/cm	8.37 ± 0.56 ^a^	7.83 ± 0.75 ^a^	8.10 ± 0.71 ^a^	7.83 ± 0.70 ^a^

All values are expressed by mean ± SD (n = 40). Values within different lowercase letters (a–d) in the same column are significantly different (*p* < 0.05) as determined by Duncan’s test.

## Data Availability

Data are contained within the article or Appendix A.

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
