# Peer review of "Effects of Pulsed Electric Field Pretreatment on Black Tea Processing and Its Impact on Cold-Brewed Tea"

_foods, 2024, doi:10.3390/foods13010164_

Round 1
Reviewer 1 Report
Comments and Suggestions for Authors
Nice paper of interesting results of PEF for tea production.
However some changes and additions have to be made:
-chapter 2.1.: As an additional important property, the typical thickness of the leaves should be given. If possible, all geometric dimensions should be given as mean plus/minus the standard deviation.
chapter 2.2: As the effect of the PEF treatments is dependent on the geometric alignment with respect to the electric field, the author should add this information.
chapter 2.3:
drying:
- what is the geometric arrangement (, are the leaves separated or do they build a layer of a certain thickness?
- what is the air humidity during drying?
infusion:
-are the leaves ground before infusion?
chapter 2.5:
-why HSB color space is used here (and in chapter 2.6. CIE lab value)
-the term "approprate) ... thresholds" must be specified, how are they chosen and what are the concrete values used?
chapter 2.6:
-please explain, how the browning index is calculated.
- l. 132: probably the mentioned 4g are the fresh mass before drying?
- eq. (2): remove the term 'x100', (only necessary for the conversion in percent)
- the equilibium moisture content has to be specified in more detail (what is the temperature and what the air humidity?)
chapter 2.9:
- the equations (4) and (5) have to be explained, where does they stem from, how can they be derived?
chapter 2.10:
- please check again: was it really only 15 ml in a 1 l-bowl?
- were the evaluators trained (and how) before the sensory evaluation?
chapter 3.1:
- concerning the mentioned 360-720 MJ/t what mass is meant here, the dry or the fresh tea leaves' mass?
- the author mention, that the 'energy requirements for PEF were lower'. Could they quantify the actual values?
chapter 3.4.2: some statements are in duplicate
chapter 3.5:
a particle size increase is not necessarily correlated with a decrease in turbidity. Is this really the reason, here?
Reviewer 2 Report
Comments and Suggestions for Authors
Comments and Recommendations to the Authors
1. The title, abstract, keywords, and introduction sections are appropriate and clear.
2. Materials and methods;
a) Line 81-82, "Fresh tea leaves were submerged in the tap water.." -- why?
b) Line 89-90, "The withering process was conducted at 23C and 60%.." -- why?
c) Perhaps a Flow chart should be included in Section 2.3 to make it easier for the readers to understand the flow of experimental works.
d) References for Sections 2.4, 2.7, 2.8, and 2.10?
e) Why sample WT12h was chosen for the sensory evaluation? Was there any statistical analysis that deduced that?
3. Result and discussion;
a) Line 196, "..solution's.." -- revise to reflect the tea leaves submerged in tap water
b) Line 215, "...was dependent ON? PEF strength.."
d) Line 210-211, "The energy requirements..." -- how much lower? any data to back this statement?
e) Line 208, :...for the rolling PROCESS.
f) Caption for Figure 3 - Please revise. It was hard to follow through especially for Figure 3-3.
g) Line 298-300, "The PEF treated leaves..." -- was there any SEM analysis included to validate this claim?
h) Section 3.4.2 - please include discussion on falling rate, constant rate and diffusion rate based on drying mechanisms.
i) Be mindful of spelling mistakes scattered in the manuscript.
4. Conclusion - ok
5. References are relevant to the study and up-to-date.
